# Calculation of Acceleration Effects Using the Zubarev Density Operator

**Georgy Prokhorov** [1,*] ⬤, **Oleg Teryaev** [1,2,3] **and Valentin Zakharov** [2,4,5]

[1] Joint Institute for Nuclear Research, 141980 Dubna, Russia; teryaev@jinr.ru
[2] Institute of Theoretical and Experimental Physics, NRC Kurchatov Institute, 117218 Moscow, Russia; vzakharov@itep.ru
[3] Department of Physics, M. V. Lomonosov Moscow State University, 117234 Moscow, Russia
[4] School of Biomedicine, Far Eastern Federal University, 690950 Vladivostok, Russia
[5] Moscow Institute of Physics and Technology, 141700 Dolgoprudny, Russia
[*] Correspondence: prokhorov@theor.jinr.ru

**Abstract:** The relativistic form of the Zubarev density operator can be used to study quantum effects associated with acceleration of the medium. In particular, it was recently shown that the calculation of perturbative corrections in acceleration based on the Zubarev density operator makes it possible to show the existence of the Unruh effect. In this paper, we present the details of the calculation of quantum correlators arising in the fourth order of the perturbation theory needed to demonstrate the Unruh effect. Expressions for the quantum corrections for massive fermions are also obtained.

**Keywords:** Zubarev operator; Unruh effect; acceleration

---

## 1. Introduction

There are wonderful quantum-field effects associated with non-uniform motion of the medium. A well-known example of such an effect is the Unruh effect, according to which an accelerated observer perceives the Minkowski vacuum as a medium filled with particles with a temperature depending on the acceleration [1]. This temperature is called the Unruh temperature, and it is equal to

$$T_U = \frac{\hbar |a|}{2\pi ck}. \tag{1}$$

The Unruh effect is similar to the Hawking effect, since it is also associated with the appearance of the event horizon in the accelerated system. This effect continues to be the focus of theorists [2–5]. The possibility of experimental observation of the Unruh effect needs the generation of ultrahigh acceleration in a system, which is relevant, in particular, for particle collisions [6,7] and systems with two-level atoms in quantum optics [8–10].

There is a universal fundamental statistical approach to describing the equilibrium thermodynamics of quantized fields. This approach is based on the relativistic form of the Zubarev density operator [11,12]. It has recently been shown that this approach allows to study in a regular way the effects of rotation and acceleration in a medium of relativistic particles [13–15].

Using the Zubarev operator method, various effects associated with the motion of the medium are shown. In particular, the well-known chiral vortical effect is shown and corrections to this effect are calculated [13,14,16]. Since the chiral vortical effect is associated with the axial electromagnetic anomaly [17–19], as well as with the gravitational anomaly [20,21], it turns out that the approach with the Zubarev operator carries information about the most fundamental properties of matter.

A remarkable observation made recently is that the Unruh effect can also be obtained from the Zubarev density operator [22,23]. Relativistic quantum statistical mechanics considers a continuous medium filled with particles described by quantized fields. This medium in equilibrium is characterized by a number of thermodynamic parameters, such as temperature, energy density, pressure, and others. Non-trivial aspect is connected with the need for normalization of the thermodynamic quantities of the system to a specific vacuum, as a rule, the Minkowski vacuum. With such a statement of the problem, a direct consequence of the Unruh effect from the point of view of quantum statistical mechanics is the vanishing of the observables, in particular, the energy-momentum tensor, at a proper temperature equal to the Unruh temperature [24,25]. This is exactly what was found in [23].

This means that, in the Zubarev approach, nontrivial gravitational effects, associated with the occurrence of an event horizon, and the changes in vacuum properties depending on the reference system, are reproduced. This observation seems even more surprising because the corresponding calculation was carried out in ordinary flat Minkowski space-time, that is, by observing an accelerated medium from an inertial frame. Nevertheless, nontrivial physics associated with Unruh effect is reproduced.

Moreover, as discussed in [26], the Zubarev density operator exactly reproduces quantum corrections that were derived in space of a cosmic string, characterized by a conical singularity [25,27]. The existence of such exact duality means that the Zubarev operator of the accelerated medium leads to emergent conical geometry.

To justify the Unruh effect in [23], it was necessary to calculate a five-point correlator with boost operators and energy-momentum tensor. This calculation in [23] was made for the massless Dirac field. The method we used was developed in a series of works [13–15], where the perturbation theory with the boost operator was developed and corrections up to the second order were calculated. It is well known [13] that the boost operator does not commute with the Hamiltonian of the system. Because of this, with each subsequent order of the perturbation theory, the complexity of calculation of the corresponding quantum correlators increases. The fourth order found in [23] is currently a record one. In the present paper, we describe a never before given scheme for calculating higher orders of the perturbation theory with the boost operator and also derive expressions for the fourth-order corrections to the energy-momentum tensor at nonzero mass.

To date, the Unruh effect has been considered from various points of view. In particular, in the framework of quantum optics [8–10], the Unruh effect manifests itself in the thermal distribution with the Unruh temperature, in the probability of absorption and emission of gamma quanta by accelerated two-level atoms. It is necessary to consider the interaction of atoms with an electromagnetic field in the framework of perturbation theory with respect to the coupling constant, while acceleration effects can be taken into account in a nonperturbative way through Rindler coordinates.

Despite the difference in approaches, a parallel can be established between our consideration and the usual approach to the Unruh effect, as well as quantum optics. In particular, in the statistical approach we also obtained a term in the energy density (which is the last term in Equation (3.1) in [23]), which corresponds to the Bose distribution of gamma quanta at the Unruh temperature.

The paper has the following structure. Section 2 introduces the basic concepts of the method of Zubarev density operator. An algorithm of constructing a perturbation theory in acceleration is also discussed. In the Section 3 we describe in details the calculation of the corrections of the fourth-order in acceleration to the energy-momentum tensor. The interpretation associated with the Unruh effect is given in Section 4. In Section 5 the conclusions are given. In the Appendix A the formulas for the coefficients at finite mass are presented.

The system of units $\hbar = c = k = 1$ is used.

## 2. Perturbation Theory in Acceleration Based on the Zubarev Density Operator

In this section, we introduce the basic concepts related to the density operator and describe how the acceleration perturbation theory can be constructed. In general, in this section we follow the paper [13]. In [11,12], a relativistic form of the Zubarev density operator was obtained for a medium in a state of local thermodynamic equilibrium

$$\hat{\rho} = \frac{1}{Z} \exp\left\{ -\int_{\Sigma} d\Sigma_{\mu} [\hat{T}^{\mu\nu}(x)\beta_{\nu}(x) - \xi(x)\hat{j}^{\mu}(x)] \right\},\tag{2}$$

where integration over the three-dimensional hypersurface $\Sigma$ is performed. Here, $\beta_{\mu} = \frac{u_{\mu}}{T}$ is the inverse temperature 4-vector, $T$ is the proper temperature, $\xi = \frac{u}{T}$ is the ratio of the chemical potential in the co-moving frame to temperature, $\hat{T}^{\mu\nu}$ and $\hat{j}^{\mu}$ are the energy-momentum tensor and current operators. The conditions of global thermodynamic equilibrium for a medium with rotation and acceleration, that is, conditions under which the density operator (2) becomes independent on the choice of the hypersurface $\Sigma$, over which the integration occurs, thus acquiring the properties of a density operator in a state of global thermodynamic equilibrium, have the form [13,15,28,29]

$$\beta_{\mu} = b_{\mu} + \varpi_{\mu\nu}x_{\nu}, \quad b_{\mu} = \text{const}, \quad \varpi_{\mu\nu} = \text{const},$$
$$\varpi_{\mu\nu} = -\frac{1}{2}(\partial_{\mu}\beta_{\nu} - \partial_{\nu}\beta_{\mu}), \quad \xi = \text{const},\tag{3}$$

where $\varpi_{\mu\nu}$ is the thermal vorticity tensor. In the general case, integration over the hypersurface is to be done and the quantum statistical theory should be projected to this hypersurface [30–32]. So under the condition (3), the density operator (2) becomes the global equilibrium density operator [13,15,22]

$$\hat{\rho} = \frac{1}{Z} \exp\left\{ -\beta_{\mu}(x)\hat{P}^{\mu} + \frac{1}{2}\varpi_{\mu\nu}\hat{J}_{x}^{\mu\nu} + \xi\hat{Q} \right\},\tag{4}$$

where $\hat{P}$ is the 4-momentum operator, $\hat{Q}$ is the charge operator, and $\hat{J}_{x}$ are the generators of the Lorentz transformations shifted to the point $x$

$$\hat{J}_{x}^{\mu\nu} = \int d\Sigma_{\lambda} \left[ (y^{\mu} - x^{\mu})\hat{T}^{\lambda\nu}(y) - (y^{\nu} - x^{\nu})\hat{T}^{\lambda\mu}(y) \right].\tag{5}$$

The technique of calculating the mean values of physical quantities based on (4) was developed in [13,15], in which second-order corrections in the thermal vorticity tensor were calculated to various thermodynamic quantities for scalar and Dirac fields.

Note that the condition (3) also lead to a system of kinematic equations of motion, solving which, we can construct trajectories of motion. Particular cases of this solution are the rotation of the medium as a solid, as well as uniformly accelerated motion.

Following [13], we introduce the thermal acceleration vector $\alpha_{\mu}$ and the thermal vorticity pseudo-vector $w_{\mu}$

$$\alpha_{\mu} = \varpi_{\mu\nu}u^{\nu}, \quad w_{\mu} = -\frac{1}{2}\epsilon_{\mu\nu\alpha\beta}u^{\nu}\varpi^{\alpha\beta}.\tag{6}$$

Drawing a parallel with the electrodynamics, $\alpha_{\mu}$ and $w_{\mu}$ can be called the "electrical" and "magnetic" components of the tensor $\varpi$. The tensor $\varpi_{\mu\nu}$ can be decomposed into these components as follows

$$\varpi_{\mu\nu} = \epsilon_{\mu\nu\alpha\beta}w^{\alpha}u^{\beta} + \alpha_{\mu}u_{\nu} - \alpha_{\nu}u_{\mu}.\tag{7}$$

The meaning of the vectors $\alpha_\mu$ and $w_\mu$ becomes clear when considering the case of global thermodynamic equilibrium, in which they are proportional to the usual kinematic 4-acceleration $a_\mu$ and vorticity $\omega_\mu$

$$\alpha^\mu = \varpi^\mu{}_\nu u^\nu = u^\nu \partial_\nu \beta^\mu = \frac{1}{T} u^\nu \partial_\nu u^\mu = \frac{a^\mu}{T} \,, \tag{8}$$

and for thermal vorticity, we get

$$w_\mu = -\frac{1}{2} \epsilon_{\mu\nu\alpha\beta} u^\nu \varpi^{\alpha\beta} = -\frac{1}{2} \epsilon_{\mu\nu\alpha\beta} u^\nu \partial^\beta \beta^\alpha = \frac{1}{2T} \epsilon_{\mu\nu\alpha\beta} u^\nu \partial^\alpha u^\beta = \frac{\omega_\mu}{T} \,. \tag{9}$$

In the rest frame, $a^\mu$ and $\omega^\mu$ are expressed in terms of three-dimensional vectors

$$a^\mu = (0, \mathbf{a}), \quad \omega^\mu = (0, \mathbf{w}) \,, \tag{10}$$

where $\mathbf{a}$ and $\mathbf{w}$ are three-dimensional acceleration and angular velocity.

The density operator (4) allows one to find corrections related to thermal vorticity in the framework of perturbation theory. To do this, it is necessary to expand (4) in a series taking into account the fact that we are constructing a perturbation theory with non-commuting operators. According to [13] we have

$$\langle \hat{O}(x) \rangle = \langle \hat{O}(0) \rangle_{\beta(x)} + \sum_{N=1}^{\infty} \frac{\varpi^N}{2^N N! |\beta|^N} \int_0^{|\beta|} d\tau_1 d\tau_2 ... d\tau_N \langle T_\tau \hat{J}_{-i\tau_1 u} ... \hat{J}_{-i\tau_N u} \hat{O}(0) \rangle_{\beta(x),c} \,, \tag{11}$$

where it is assumed that each of the thermal vorticity tensors is contracted with the tensor $\hat{J}$ so that $\varpi_{\mu\nu} \hat{J}^{\mu\nu}$. Equation (11) includes only connected correlators, all disconnected correlators are reduced due to the contribution of the denominator $1/Z$ to (4). This fact is reflected in the subscript $c$; the subscript $\beta(x)$ means that the mean values are taken at $\varpi = 0$, that is, averaging is performed over a grand canonical distribution. The $T_\tau$ operator orders in imaginary time $\tau$, and $|\beta| = \sqrt{\beta^\mu \beta_\mu} = \frac{1}{T}$.

It is convenient to introduce the boost operator $\hat{K}$ and the angular momentum operator $\hat{J}$

$$\hat{J}^{\mu\nu} = u^\mu \hat{K}^\nu - u^\nu \hat{K}^\mu - \epsilon^{\mu\nu\rho\sigma} u_\rho \hat{J}_\sigma \,. \tag{12}$$

From (7) and (12), it follows that scalar products with vorticity tensor in (4) and (11) decompose into terms with boost and angular momentum

$$\varpi_{\mu\nu} \hat{J}_x^{\mu\nu} = -2\alpha_\mu \hat{K}_x^\mu - 2w_\mu \hat{J}_x^\mu \,. \tag{13}$$

Further, we will consider uniformly accelerated media without vorticity and chemical potential; therefore (4), transforms to the density operator of the form

$$\hat{\rho} = \frac{1}{Z} \exp \left\{ -\beta_\mu \hat{P}^\mu - \alpha_\mu \hat{K}_x^\mu \right\} \,, \tag{14}$$

and the perturbation theory in (11) takes the form of the series in acceleration

$$\langle \hat{O}(x) \rangle = \langle \hat{O}(0) \rangle_{\beta(x)} + \sum_{N=1}^{\infty} \frac{(-1)^N a^N}{N!} \int_0^{|\beta|} d\tau_1 d\tau_2 ... d\tau_N \langle T_\tau \hat{K}_{-i\tau_1 u} ... \hat{K}_{-i\tau_N u} \hat{O}(0) \rangle_{\beta(x),c} \,, \tag{15}$$

## 3. Calculation of Fourth-Order Coefficients in Acceleration

The second-order coefficients in acceleration in the energy-momentum tensor of the Dirac field were calculated in [13,14]. In this section, we present the details of calculation of the fourth-order coefficient.

The operator form of the energy-momentum tensor of the mass-less Dirac fields is well known. We will use the symmetrized Belinfante energy-momentum tensor

$$\hat{T}^{\mu\nu} = \frac{i}{4} \left( \overline{\Psi}\gamma^{\mu}\partial^{\nu}\Psi - \partial^{\nu}\overline{\Psi}\gamma^{\mu}\Psi + \overline{\Psi}\gamma^{\nu}\partial^{\mu}\Psi - \partial^{\mu}\overline{\Psi}\gamma^{\nu}\Psi \right) . \tag{16}$$

As follows from (15), the calculation of the necessary correlators is performed in imaginary time—a time shift is made along the imaginary axis. Thus, it is necessary to pass to the Euclidean formalism in imaginary time. The Euclidean version of the energy-momentum tensor (16) has the form

$$\hat{T}^{\mu\nu} = \frac{i^{\delta_{0\mu}+\delta_{0\nu}}}{4} \left( \overline{\Psi}\tilde{\gamma}^{\mu}\partial^{\nu}\Psi - \partial^{\nu}\overline{\Psi}\tilde{\gamma}^{\mu}\Psi + \overline{\Psi}\tilde{\gamma}^{\nu}\partial^{\mu}\Psi - \partial^{\mu}\overline{\Psi}\tilde{\gamma}^{\nu}\Psi \right) , \tag{17}$$

where $\tilde{\gamma}$ are the Euclidean Dirac matrices

$$\tilde{\gamma}_{\mu} = i^{1-\delta_{0\mu}}\gamma_{\mu} , \quad \tilde{\gamma}^{\mu} = i^{1-\delta_{0\mu}}\gamma^{\mu} , \quad \{\tilde{\gamma}_{\mu}\tilde{\gamma}_{\nu}\} = 2\delta_{\mu\nu} , \tag{18}$$

and derivatives are also taken in Euclidean space-time, so that

$$\tilde{\partial}_{\mu} = (-i)^{\delta_{0\mu}}\partial_{\mu} . \tag{19}$$

However, we will omit the tilde sign for derivatives. Consider the mean value of the energy-momentum tensor in the fourth order of the perturbation theory in acceleration using (15)

$$\begin{aligned} \langle \hat{T}^{\mu\nu}(x) \rangle &= \langle \hat{T}^{\mu\nu}(0) \rangle_{\beta(x)} + \frac{a_{\rho}a_{\sigma}}{2} \int_0^{|\beta|} d\tau_1 d\tau_2 \langle T_{\tau}\hat{K}^{\rho}_{-i\tau_1 u}\hat{K}^{\sigma}_{-i\tau_2 u}\hat{T}^{\mu\nu}(0) \rangle_{\beta(x),c} \\ &\quad + \frac{8a_{\rho}a_{\sigma}a_{\gamma}a_{\eta}}{4!} \int_0^{|\beta|} d\tau_1 d\tau_2 d\tau_3 d\tau_4 \langle T_{\tau}\hat{K}^{\rho}_{-i\tau_1 u}\hat{K}^{\sigma}_{-i\tau_2 u}\hat{K}^{\gamma}_{-i\tau_3 u}\hat{K}^{\eta}_{-i\tau_4 u}\hat{T}^{\mu\nu}(0) \rangle_{\beta(x),c} + \mathcal{O}(a^6) . \end{aligned} \tag{20}$$

Symmetry and parity considerations fix the form of the energy-momentum tensor in the fourth order of perturbation theory

$$\begin{aligned} \langle \hat{T}^{\mu\nu} \rangle &= (\rho_0 - A_1 T^2 a^2 + A_2 a^4)u^{\mu}u^{\nu} - (p_0 - A_3 T^2 a^2 + A_4 a^4)\Delta^{\mu\nu} \\ &\quad + (A_5 T^2 - A_6 a^2)a^{\mu}a^{\nu} + \mathcal{O}(a^6) \qquad \Delta^{\mu\nu} = g^{\mu\nu} - u^{\mu}u^{\nu} , \end{aligned} \tag{21}$$

where $a^2 = a_{\mu}a^{\mu}$. As already mentioned, 2-order coefficients were calculated earlier in [13,14]. Our goal is to calculate coefficients of the 4th order $A_2$, $A_4$, $A_6$. Comparing (20) with (21), we obtain

$$\begin{aligned} A_2 a^4 u^{\mu}u^{\nu} - A_4 a^4 \Delta^{\mu\nu} - A_6 a^2 a^{\mu}a^{\nu} &= \frac{a_{\rho}a_{\sigma}a_{\gamma}a_{\eta}}{4!} \int_0^{|\beta|} d\tau_1 d\tau_2 d\tau_3 d\tau_4 \\ &\quad \times \langle T_{\tau}\hat{K}^{\rho}_{-i\tau_1 u}\hat{K}^{\sigma}_{-i\tau_2 u}\hat{K}^{\gamma}_{-i\tau_3 u}\hat{K}^{\eta}_{-i\tau_4 u}\hat{T}^{\mu\nu}(0) \rangle_{\beta(x),c} . \end{aligned} \tag{22}$$

The coefficients $A_2, A_4, A_6$ are Lorentz invariants, and the relation (22) is valid for any choice of the vectors $u_\mu, a_\mu$. Therefore, to determine the coefficient, we can choose the vectors $u_\mu, a_\mu$ in any form convenient for us. To determine $A_2$, we choose $a^\mu = (0,0,0,|a|)$ and $u^\mu = (1,0,0,0)$ and consider the components $\mu = 0, \nu = 0$, to determine $A_4$ we choose $a^\mu = (0,0,|a|,0)$ and $u^\mu = (1,0,0,0)$ and consider the components $\mu = 3, \nu = 3$, and to determine $A_6$ we choose $a^\mu = (0,0,0,|a|)$ and $u^\mu = (1,0,0,0)$ and consider the components $\mu = 3, \nu = 3$. As a result, we obtain

$$
\begin{aligned}
A_2 &= \frac{1}{4!} \int_0^{|\beta|} d\tau_1 d\tau_2 d\tau_3 d\tau_4 \langle T_\tau \hat{K}^3_{-i\tau_1 u} \hat{K}^3_{-i\tau_2 u} \hat{K}^3_{-i\tau_3 u} \hat{K}^3_{-i\tau_4 u} \hat{T}^{00}(0) \rangle_{\beta(x),c}, \\
A_4 &= \frac{1}{4!} \int_0^{|\beta|} d\tau_1 d\tau_2 d\tau_3 d\tau_4 \langle T_\tau \hat{K}^2_{-i\tau_1 u} \hat{K}^2_{-i\tau_2 u} \hat{K}^2_{-i\tau_3 u} \hat{K}^2_{-i\tau_4 u} \hat{T}^{33}(0) \rangle_{\beta(x),c}, \\
A_6 &= -A_4 + \frac{1}{4!} \int_0^{|\beta|} d\tau_1 d\tau_2 d\tau_3 d\tau_4 \langle T_\tau \hat{K}^3_{-i\tau_1 u} \hat{K}^3_{-i\tau_2 u} \hat{K}^3_{-i\tau_3 u} \hat{K}^3_{-i\tau_4 u} \hat{T}^{33}(0) \rangle_{\beta(x),c}.
\end{aligned}
\tag{23}
$$

We now use the representation of the boost operator through the energy-momentum tensor. According to (5) and (12), we have

$$
\begin{aligned}
\hat{K}^3_{-i\tau u} &= \hat{J}^{03}_{-i\tau u} = \int d^3 x (-1) x^3 \hat{T}^{00}(\tau, \mathbf{x}), \\
\hat{K}^2_{-i\tau u} &= \hat{J}^{02}_{-i\tau u} = \int d^3 x (-1) x^2 \hat{T}^{00}(\tau, \mathbf{x}),
\end{aligned}
\tag{24}
$$

Substituting (24) into (23), we come to the need of calculating quantities of the form

$$
\begin{aligned}
C^{\alpha_1\alpha_2|\alpha_3\alpha_4|\alpha_5\alpha_6|\alpha_7\alpha_8|\alpha_9\alpha_{10}|ijkl} &= \int_0^{|\beta|} d\tau_x d\tau_y d\tau_z d\tau_f d^3 x d^3 y d^3 z d^3 f \\
&\times x^i y^j z^k f^l \langle T_\tau \hat{T}^{\alpha_1\alpha_2}(\tau_x, \mathbf{x}) \hat{T}^{\alpha_3\alpha_4}(\tau_y, \mathbf{y}) \hat{T}^{\alpha_5\alpha_6}(\tau_z, \mathbf{z}) \hat{T}^{\alpha_7\alpha_8}(\tau_f, \mathbf{f}) \hat{T}^{\alpha_9\alpha_{10}}(0) \rangle_{\beta(x),c}.
\end{aligned}
\tag{25}
$$

In particular, from (23), we have

$$
A_2 = \frac{1}{4!} C^{00|00|00|00|00|3333}, \quad A_4 = \frac{1}{4!} C^{00|00|00|00|33|2222}, \quad A_6 = -A_4 + \frac{1}{4!} C^{00|00|00|00|33|3333}.
\tag{26}
$$

Next, we will focus on calculating the coefficient in energy $A_2$; the remaining coefficients can be calculated by analogy.

We represent the energy-momentum tensor (17) in a split form

$$
\begin{aligned}
\hat{T}^{\alpha\beta}(X) &= \lim_{X_1, X_2 \to X} \mathcal{D}^{\alpha\beta}_{ab}(\partial_{X_1}, \partial_{X_2}) \bar{\Psi}_a(X_1) \Psi_b(X_2), \\
\mathcal{D}^{\alpha\beta}_{ab}(\partial_{X_1}, \partial_{X_2}) &= \frac{i^{\delta_{0\alpha}+\delta_{0\beta}}}{4} [\tilde{\gamma}^\alpha_{ab}(\partial_{X_2} - \partial_{X_1})^\beta + \tilde{\gamma}^\beta_{ab}(\partial_{X_2} - \partial_{X_1})^\alpha],
\end{aligned}
\tag{27}
$$

and substitute it in (26). As a result, we get for the corresponding correlator

$$
\langle T_\tau \hat{T}^{00}(X) \hat{T}^{00}(Y) \hat{T}^{00}(Z) \hat{T}^{00}(F) \hat{T}^{00}(0) \rangle_{\beta(x),c} = \lim_{\substack{X_1, X_2 \to X \\ Y_1, Y_2 \to Y \\ Z_1, Z_2 \to Z \\ F_1, F_2 \to F \\ H_1, H_2 \to H = 0}} \mathcal{D}^{00}_{a_1 a_2}(\partial_{X_1}, \partial_{X_2})
$$

$$
\begin{aligned}
&\mathcal{D}^{00}_{a_3 a_4}(\partial_{Y_1}, \partial_{Y_2}) \mathcal{D}^{00}_{a_5 a_6}(\partial_{Z_1}, \partial_{Z_2}) \mathcal{D}^{00}_{a_7 a_8}(\partial_{F_1}, \partial_{F_2}) \mathcal{D}^{00}_{a_9 a_{10}}(\partial_{H_1}, \partial_{H_2}) \langle T_\tau \bar{\Psi}_{a_1}(X_1) \Psi_{a_2}(X_2) \\
&\times \bar{\Psi}_{a_3}(Y_1) \Psi_{a_4}(Y_2) \bar{\Psi}_{a_5}(Z_1) \Psi_{a_6}(Z_2) \bar{\Psi}_{a_7}(F_1) \Psi_{a_8}(F_2) \bar{\Psi}_{a_9}(H_1) \Psi_{a_{10}}(H_2) \rangle_{\beta(x),c}.
\end{aligned}
\tag{28}
$$

When calculating the correlator with 10 Dirac fields of the form (28), it is necessary to use an analogue of Wick theorem for field theory at finite temperatures. Then, the five-point correlator in (28) leads to the product of mean values of quadratic combinations of Dirac fields, that is, thermal propagators. For short, we denote $\Psi_{a_n} \to n$, and $\overline{\Psi}_{a_n} \to \bar{n}$ and omit $T_\tau$ and the index $\beta(x)$. Then, after extraction on the connected part in (28) according to Wick theorem, we obtain 24 terms

$$\langle T_\tau \overline{\Psi}_{a_1}(X_1)\Psi_{a_2}(X_2)\overline{\Psi}_{a_3}(Y_1)\Psi_{a_4}(Y_2)\overline{\Psi}_{a_5}(Z_1)\Psi_{a_6}(Z_2)\overline{\Psi}_{a_7}(F_1)\Psi_{a_8}(F_2)$$

$$\overline{\Psi}_{a_9}(H_1)\Psi_{a_{10}}(H_2)\rangle_{\beta(x),c} = \langle\bar{1}2\bar{3}4\bar{5}6\bar{7}8\bar{9}10\rangle = -\langle\bar{1}4\rangle\langle2\bar{9}\rangle\langle\bar{3}6\rangle\langle\bar{5}8\rangle\langle\bar{7}10\rangle$$

$$+\langle\bar{1}4\rangle\langle2\bar{7}\rangle\langle\bar{3}6\rangle\langle\bar{5}10\rangle\langle8\bar{9}\rangle + \langle\bar{1}4\rangle\langle2\bar{9}\rangle\langle\bar{3}8\rangle\langle\bar{5}10\rangle\langle6\bar{7}\rangle + \langle\bar{1}4\rangle\langle2\bar{5}\rangle\langle\bar{3}8\rangle\langle6\bar{9}\rangle\langle\bar{7}10\rangle$$

$$+\langle\bar{1}4\rangle\langle2\bar{7}\rangle\langle\bar{3}10\rangle\langle\bar{5}8\rangle\langle6\bar{9}\rangle - \langle\bar{1}4\rangle\langle2\bar{5}\rangle\langle\bar{3}10\rangle\langle6\bar{7}\rangle\langle8\bar{9}\rangle + \langle\bar{1}6\rangle\langle2\bar{9}\rangle\langle\bar{3}8\rangle\langle4\bar{5}\rangle\langle\bar{7}10\rangle$$

$$-\langle\bar{1}6\rangle\langle2\bar{7}\rangle\langle\bar{3}10\rangle\langle4\bar{5}\rangle\langle8\bar{9}\rangle + \langle\bar{1}6\rangle\langle2\bar{9}\rangle\langle\bar{3}10\rangle\langle4\bar{7}\rangle\langle\bar{5}8\rangle + \langle\bar{1}6\rangle\langle2\bar{3}\rangle\langle4\bar{9}\rangle\langle\bar{5}8\rangle\langle\bar{7}10\rangle$$

$$+\langle\bar{1}6\rangle\langle2\bar{7}\rangle\langle\bar{3}8\rangle\langle4\bar{9}\rangle\langle\bar{5}10\rangle - \langle\bar{1}6\rangle\langle2\bar{3}\rangle\langle4\bar{7}\rangle\langle\bar{5}10\rangle\langle8\bar{9}\rangle + \langle\bar{1}8\rangle\langle2\bar{9}\rangle\langle\bar{3}6\rangle\langle4\bar{7}\rangle\langle\bar{5}10\rangle$$

$$-\langle\bar{1}8\rangle\langle2\bar{5}\rangle\langle\bar{3}10\rangle\langle4\bar{7}\rangle\langle6\bar{9}\rangle - \langle\bar{1}8\rangle\langle2\bar{9}\rangle\langle\bar{3}10\rangle\langle4\bar{5}\rangle\langle6\bar{7}\rangle - \langle\bar{1}8\rangle\langle2\bar{3}\rangle\langle4\bar{9}\rangle\langle\bar{5}10\rangle\langle6\bar{7}\rangle$$

$$+\langle\bar{1}8\rangle\langle2\bar{5}\rangle\langle\bar{3}6\rangle\langle4\bar{9}\rangle\langle\bar{7}10\rangle - \langle\bar{1}8\rangle\langle2\bar{3}\rangle\langle4\bar{5}\rangle\langle6\bar{9}\rangle\langle\bar{7}10\rangle + \langle\bar{1}10\rangle\langle2\bar{7}\rangle\langle\bar{3}6\rangle\langle4\bar{9}\rangle\langle\bar{5}8\rangle$$

$$-\langle\bar{1}10\rangle\langle2\bar{5}\rangle\langle\bar{3}8\rangle\langle4\bar{9}\rangle\langle6\bar{7}\rangle - \langle\bar{1}10\rangle\langle2\bar{7}\rangle\langle\bar{3}8\rangle\langle4\bar{5}\rangle\langle6\bar{9}\rangle - \langle\bar{1}10\rangle\langle2\bar{3}\rangle\langle4\bar{7}\rangle\langle\bar{5}8\rangle\langle6\bar{9}\rangle$$

$$-\langle\bar{1}10\rangle\langle2\bar{5}\rangle\langle\bar{3}6\rangle\langle4\bar{7}\rangle\langle8\bar{9}\rangle + \langle\bar{1}10\rangle\langle2\bar{3}\rangle\langle4\bar{5}\rangle\langle6\bar{7}\rangle\langle8\bar{9}\rangle , \tag{29}$$

where signs correspond to the number of permutations of anti-commuting fields. Thermal propagators have a standard form [13,33]

$$G_{a_1 a_2}(X_1, X_2) = \langle T_\tau \Psi_{a_1}(X_1)\overline{\Psi}_{a_2}(X_2)\rangle_{\beta(x)} = \sum_P \!\!\!\!\!\!\!\int e^{iP^+(X_1-X_2)}(-iP_\mu^+\tilde{\gamma}_\mu + m)_{a_1 a_2}\Delta(P^+) ,$$

$$\bar{G}_{a_1 a_2}(X_1, X_2) = \langle T_\tau \overline{\Psi}_{a_1}(X_1)\Psi_{a_2}(X_2)\rangle_{\beta(x)} = -\langle T_\tau \Psi_{a_2}(X_2)\overline{\Psi}_{a_1}(X_1)\rangle_{\beta(x)}$$

$$= -\sum_P \!\!\!\!\!\!\!\int e^{iP^-(X_1-X_2)}(iP_\mu^-\tilde{\gamma}_\mu + m)_{a_2 a_1}\Delta(P^-) , \tag{30}$$

where integration over the three-dimensional components of the momentum and summation over the Matsubara frequencies of fermion field appear. The following notations are used in (30): $P^\pm = (p_n^\pm, \mathbf{p})$, $p_n^\pm = \pi(2n+1)/|\beta| \pm i\mu$, $n = 0, \pm 1, \pm 2, \cdots$, $X = (\tau, \mathbf{x})$, $\sum_P \!\!\!\!\int = \frac{1}{|\beta|}\sum_{n=-\infty}^{\infty}\int\frac{d^3p}{(2\pi)^3}$, and $\Delta(P) = \frac{1}{P^2+m^2}$, where the square is taken with the Euclidean metric, as also in $P_\mu^\pm\tilde{\gamma}_\mu = \not{P}^\pm$ (unlike $P^+(X_1 - X_2)$ according to [33]). Since we consider mass-less field at zero chemical potential, the mass and chemical potential must be set equal to zero $m = 0, \mu = 0$. Nevertheless, we retain the notation $P^\pm$, bearing in mind the possibility of generalization to the case with nonzero chemical potential in the future.

Next, substitute (30) in (29). We will describe the calculations for the first term in (29), while all other terms can be calculated by analogy

$$- \lim_{\substack{X_1,X_2 \to X \\ Y_1,Y_2 \to Y \\ Z_1,Z_2 \to Z \\ F_1,F_2 \to F \\ H_1,H_2 \to H=0}} \mathcal{D}^{00}_{a_1 a_2}(\partial_{X_1}, \partial_{X_2}) \mathcal{D}^{00}_{a_3 a_4}(\partial_{Y_1}, \partial_{Y_2}) \mathcal{D}^{00}_{a_5 a_6}(\partial_{Z_1}, \partial_{Z_2}) \mathcal{D}^{00}_{a_7 a_8}(\partial_{F_1}, \partial_{F_2}) \mathcal{D}^{00}_{a_9 a_{10}}(\partial_{H_1}, \partial_{H_2})$$

$$\times \bar{G}_{a_1 a_4}(X_1, Y_2) G_{a_2 a_9}(X_2, H_1) \bar{G}_{a_3 a_6}(Y_1, Z_2) \bar{G}_{a_5 a_8}(Z_1, F_2) \bar{G}_{a_7 a_{10}}(F_1, H_2) =$$

$$- \sum_{\{P,Q,K,R,L\}} \oint e^{-i\mathbf{p}(\mathbf{x}-\mathbf{y}) - i\mathbf{q}\mathbf{x} - i\mathbf{k}(\mathbf{y}-\mathbf{z}) - i\mathbf{r}(\mathbf{z}-\mathbf{f}) - i\mathbf{l}\mathbf{f}} e^{i p_n^-(\tau_x - \tau_y) + i q_n^+ \tau_x + i k_n^-(\tau_y - \tau_z) + i r_n^-(\tau_z - \tau_f) + i l_n^- \tau_f}$$

$$\times \Delta(P^-)\Delta(Q^+)\Delta(K^-)\Delta(R^-)\Delta(L^-)$$

$$\times \mathrm{tr}\Big[ (-i\rlap{L}{\,/}^-) \mathcal{D}^{00}(iL^-, -iR^-)(-i\rlap{R}{\,/}^-) \mathcal{D}^{00}(iR^-, -iK^-)(-i\rlap{K}{\,/}^-)$$

$$\times \mathcal{D}^{00}(iK^-, -iP^-)(-i\rlap{P}{\,/}^-) \mathcal{D}^{00}(iP^-, iQ^+)(-i\rlap{Q}{\,/}^+) \mathcal{D}^{00}(-iQ^+, -iL^-) \Big], \tag{31}$$

where it was necessary to arrange all the matrices under the trace in accordance with the order of the spinor indices. To calculate (31), it is necessary to find a trace of the form

$$\mathrm{tr}\Big[ \rlap{P}{\,/}_1 \mathcal{D}^{00}(P_2, P_3) \rlap{P}{\,/}_4 \mathcal{D}^{00}(P_5, P_6) \rlap{P}{\,/}_7 \mathcal{D}^{00}(P_8, P_9) \rlap{P}{\,/}_{10} \mathcal{D}^{00}(P_{11}, P_{12}) \rlap{P}{\,/}_{13} \mathcal{D}^{00}(P_{14}, P_{15}) \Big]. \tag{32}$$

The subsequent calculations are more convenient to carry out using special software applications. Calculation (32) requires finding the trace of 10 Euclidean Dirac matrices

$$\mathrm{tr}\Big[ \tilde{\gamma}^{\alpha_1} \tilde{\gamma}^{\alpha_2} \tilde{\gamma}^{\alpha_3} \tilde{\gamma}^{\alpha_4} \tilde{\gamma}^{\alpha_5} \tilde{\gamma}^{\alpha_6} \tilde{\gamma}^{\alpha_7} \tilde{\gamma}^{\alpha_8} \tilde{\gamma}^{\alpha_9} \tilde{\gamma}^{\alpha_{10}} \Big]. \tag{33}$$

Using the definition (18), this trace can be easily transformed to the trace of ordinary Dirac matrices, which can be found using standard methods. We denote the trace in (32) as $A(P_1, P_2, P_3, P_4, P_5, P_6, P_7, P_8, P_9, P_{10}, P_{11}, P_{12}, P_{13}, P_{14}, P_{15})$. Then (31) will be presented in the form

$$- \int \frac{d^3 p\, d^3 q\, d^3 k\, d^3 r\, d^3 l}{(2\pi)^{15}} e^{-i\mathbf{p}(\mathbf{x}-\mathbf{y}) - i\mathbf{q}\mathbf{x} - i\mathbf{k}(\mathbf{y}-\mathbf{z}) - i\mathbf{r}(\mathbf{z}-\mathbf{f}) - i\mathbf{l}\mathbf{f}}$$

$$\times \sum_{p_n, q_n, k_n, r_n, l_n} \frac{1}{|\beta|^5} e^{i p_n^-(\tau_x - \tau_y) + i q_n^+ \tau_x + i k_n^-(\tau_y - \tau_z) + i r_n^-(\tau_z - \tau_f) + i l_n^- \tau_f}$$

$$\times \Delta(P^-)\Delta(Q^+)\Delta(K^-)\Delta(R^-)\Delta(L^-)$$

$$\times A(-iL^-, iL^-, -iR^-, -iR^-, iR^-, -iK^-, -iK^-, iK^-, -iP^-, -iP^-, iP^-, iQ^+,$$

$$-iQ^+, -iQ^+, -iL^-). \tag{34}$$

Next, one needs to sum over the Matsubara frequencies in (34) using the relation

$$\frac{1}{|\beta|} \sum_{\omega_n} \frac{(\omega_n \pm i\mu)^k e^{i(\omega_n \pm i\mu)\tau}}{(\omega_n \pm i\mu)^2 + E^2} = \frac{1}{2E} \sum_{s=\pm 1} (-isE)^k e^{\tau sE} [\theta(-s\tau) - n_F(E \pm s\mu)], \tag{35}$$

where $E = \sqrt{\mathbf{p}^2 + m^2}$, $n_F(E) = 1/(1 + e^{E/T})$ is the Fermi distribution, and $\theta$ is the Heaviside theta function. Again, we can take $m = 0$, $\mu = 0$. As a result, we obtain

$$-\int \frac{d^3p\,d^3q\,d^3k\,d^3r\,d^3l}{(2\pi)^{15}} e^{-i\mathbf{p}(\mathbf{x}-\mathbf{y})-i\mathbf{q}\mathbf{x}-i\mathbf{k}(\mathbf{y}-\mathbf{z})-i\mathbf{r}(\mathbf{z}-\mathbf{f})-i\mathbf{l}\mathbf{f}}$$

$$\times \frac{1}{32E_pE_qE_kE_rE_l} \sum_{s_1,s_2,s_3,s_4,s_5} e^{(\tau_x-\tau_y)s_1E_p+\tau_xs_2E_q+(\tau_y-\tau_z)s_3E_k+(\tau_z-\tau_f)s_4E_r+\tau_fs_5E_l}$$

$$\times A(-i\widetilde{L},i\widetilde{L},-i\widetilde{R},-i\widetilde{R},i\widetilde{R},-i\widetilde{K},-i\widetilde{K},i\widetilde{K},-i\widetilde{P},-i\widetilde{P},i\widetilde{P},i\widetilde{Q},-i\widetilde{Q},-i\widetilde{Q},-i\widetilde{L})$$

$$\times \left(\theta\left[-s_1(\tau_x-\tau_y)\right]-n_F\left(E_p\right)\right)\left(\theta\left[-s_2\right]-n_F\left(E_q\right)\right)$$

$$\times \left(\theta\left[-s_3(\tau_y-\tau_z)\right]-n_F\left(E_k\right)\right)\left(\theta\left[-s_4(\tau_z-\tau_f)\right]-n_F\left(E_r\right)\right)\left(\theta\left[-s_5\right]-n_F\left(E_l\right)\right). \tag{36}$$

Here, following [13], the notations $\widetilde{P} = \widetilde{P}(s_1) = (-is_1E_p, \mathbf{p})$, $\widetilde{Q} = \widetilde{Q}(s_2), \cdots$ are introduced. We return now to the formula for $A_2$ (26) with spatial integrals and calculate the contribution of the term (36). This contribution has the form

$$A_2 = \int \frac{d\tau_x d\tau_y d\tau_z d\tau_f d^3x\,d^3y\,d^3z\,d^3f\,d^3p\,d^3q\,d^3k\,d^3r\,d^3l}{4!(2\pi)^{15}} e^{-i\mathbf{p}(\mathbf{x}-\mathbf{y})-i\mathbf{q}\mathbf{x}-i\mathbf{k}(\mathbf{y}-\mathbf{z})-i\mathbf{r}(\mathbf{z}-\mathbf{f})-i\mathbf{l}\mathbf{f}}$$

$$\times x^3y^3z^3f^3D + \cdots, \tag{37}$$

where the ellipsis indicates the contribution of the remaining 23 terms from (29), and $D$ equals to

$$D = -\frac{1}{32E_pE_qE_kE_rE_l} \sum_{s_1,s_2,s_3,s_4,s_5} e^{(\tau_x-\tau_y)s_1E_p+\tau_xs_2E_q+(\tau_y-\tau_z)s_3E_k+(\tau_z-\tau_f)s_4E_r+\tau_fs_5E_l}$$

$$\times A(-i\widetilde{L},i\widetilde{L},-i\widetilde{R},-i\widetilde{R},i\widetilde{R},-i\widetilde{K},-i\widetilde{K},i\widetilde{K},-i\widetilde{P},-i\widetilde{P},i\widetilde{P},i\widetilde{Q},-i\widetilde{Q},-i\widetilde{Q},-i\widetilde{L})$$

$$\times \left(\theta\left[-s_1(\tau_x-\tau_y)\right]-n_F\left(E_p\right)\right)\left(\theta\left[-s_2\right]-n_F\left(E_q\right)\right)$$

$$\times \left(\theta\left[-s_3(\tau_y-\tau_z)\right]-n_F\left(E_k\right)\right)\left(\theta\left[-s_4(\tau_z-\tau_f)\right]-n_F\left(E_r\right)\right)\left(\theta\left[-s_5\right]-n_F\left(E_l\right)\right). \tag{38}$$

Next, one needs to rewrite the product of spatial coordinates in the integral through derivatives using the formula

$$\int d^3p\,d^3q\,d^3k\,d^3r\,d^3x\,d^3y\,d^3z\,d^3f\, F(\mathbf{p},\mathbf{q},\mathbf{k},\mathbf{r},\mathbf{l})e^{-i\mathbf{p}(\mathbf{x}-\mathbf{y})-i\mathbf{q}\mathbf{x}-i\mathbf{k}(\mathbf{y}-\mathbf{z})-i\mathbf{r}(\mathbf{z}-\mathbf{f})-i\mathbf{l}\mathbf{f}}x^3y^3z^3f^3$$

$$= (2\pi)^{12} \int d^3p \left(-\frac{\partial^3}{\partial q^3\partial l^3\partial p^3\partial r^3} - \frac{\partial^3}{\partial q^3\partial l^3\partial p^3\partial l^3}\right.$$

$$\left.+\frac{\partial^3}{\partial q^3\partial l^3\partial r^3\partial q^3} + \frac{\partial^3}{\partial q^3\partial q^3\partial l^3\partial l^3}\right) F(\mathbf{p},\mathbf{q},\mathbf{k},\mathbf{r},\mathbf{l})\Bigg|_{\substack{\mathbf{l}=\mathbf{p}\\\mathbf{r}=\mathbf{p}\\\mathbf{k}=\mathbf{p}\\\mathbf{q}=-\mathbf{p}}}, \tag{39}$$

resulting from integration by parts and properties of the delta function. After that, (37) is converted to the form

$$A_2 = \frac{1}{4!(2\pi)^3} \int d\tau_x d\tau_y d\tau_z d\tau_f d^3p \left(-\frac{\partial^3}{\partial q^3\partial l^3\partial p^3\partial r^3} - \frac{\partial^3}{\partial q^3\partial l^3\partial p^3\partial l^3}\right.$$

$$\left.+\frac{\partial^3}{\partial q^3\partial l^3\partial r^3\partial q^3} + \frac{\partial^3}{\partial q^3\partial q^3\partial l^3\partial l^3}\right) D(\mathbf{p},\mathbf{q},\mathbf{k},\mathbf{r},\mathbf{l})\Bigg|_{\substack{\mathbf{l}=\mathbf{p}\\\mathbf{r}=\mathbf{p}\\\mathbf{k}=\mathbf{p}\\\mathbf{q}=-\mathbf{p}}} + \cdots. \tag{40}$$

Now, it remains to integrate over the imaginary time and also over the last momentum, which can be done directly in spherical coordinates $d^3p = |\mathbf{p}|^2 d|\mathbf{p}| \sin(\theta) d\theta d\phi$. The sequence of actions in this case, from the point of view of calculation speed, will be most convenient as follows: first one needs to make differentiations with respect to the four momentum variables in (40), then make the corresponding changes of the variables following from the delta functions, then sum over the indices $s_n$ from (38), then integrate over the angles in $d^3p$, and then integrate over imaginary time variables, which requires careful handling of theta functions. The transformations with each of the 24 terms in (29) can be performed independently and using parallel computing tools. We do not give the described intermediate steps, since they are most conveniently performed using the program, and the intermediate formulas themselves are extremely long, while the calculations themselves are not difficult from a mathematical point of view and are done directly. As a result, we obtain the following integral

$$
\begin{aligned}
A_2 &= \int_0^\infty d\tilde{p}\, e^{\frac{9\tilde{p}}{2}} \tilde{p}^3 \left( 5600\tilde{p}\left(49\tilde{p}^2 - 95\right) \cosh\left(\frac{\tilde{p}}{2}\right) + 2016\tilde{p}\left(25 - 119\tilde{p}^2\right) \cosh\left(\frac{3\tilde{p}}{2}\right) \right. \\
&\quad + 53200\left(\sinh\left(\frac{3\tilde{p}}{2}\right) - 11\sinh\left(\frac{\tilde{p}}{2}\right)\right)\cosh^4\left(\frac{\tilde{p}}{2}\right) + \tilde{p}\left(-224\left(\tilde{p}^2 + 25\right)\cosh\left(\frac{7\tilde{p}}{2}\right)\right. \\
&\quad + 224\left(119\tilde{p}^2 + 575\right)\cosh\left(\frac{5\tilde{p}}{2}\right) + 18\tilde{p}\sinh\left(\frac{\tilde{p}}{2}\right)\left(-5786\tilde{p}^2 + \left(\tilde{p}^2 + 210\right)\cosh(3\tilde{p})\right. \\
&\quad - 6\left(41\tilde{p}^2 + 1890\right)\cosh(2\tilde{p}) + 3\left(1349\tilde{p}^2 + 9450\right)\cosh(\tilde{p}) \\
&\quad \left.\left.\left. + 39900\right)\right)\right)\left(50400\pi^2\left(e^{\tilde{p}} + 1\right)^9\right)^{-1},
\end{aligned}
$$

(41)

where the dimensionless variable $\tilde{p} = |\mathbf{p}|/T$ was introduced. This integral converges and can be found analytically:

$$
A_2 = -\frac{17}{960\pi^2}.
$$

(42)

Repeating the entire calculation algorithm for the coefficients $A_4$, $A_6$ in (26), we obtain at $m = 0$

$$
A_4 = -\frac{17}{2880\pi^2}, \quad A_6 = 0.
$$

(43)

Saving the mass in all formulas, in particular, in the propagators (30), we get more complicated expressions for the coefficients at finite mass given in the Appendix A.

## 4. Discussion

In the previous section, we described the details of the calculation of the corrections of the fourth order in acceleration to the energy-momentum tensor of the Dirac field, first obtained in [23]. Taking into account (42) and (43), we obtain the next formula for the energy-momentum tensor at $m = 0$

$$
\langle \hat{T}^{\mu\nu} \rangle = \left(\frac{7\pi^2 T^4}{60} + \frac{T^2|a|^2}{24} - \frac{17|a|^4}{960\pi^2}\right)u^\mu u^\nu - \left(\frac{7\pi^2 T^4}{180} + \frac{T^2|a|^2}{72} - \frac{17|a|^4}{2880\pi^2}\right)\Delta^{\mu\nu} + \mathcal{O}(a^6),
$$

(44)

where the notation $|a| = \sqrt{-a_\mu a^\mu}$ is used.

As discussed in [22,24,25,27], the mean value of the energy-momentum tensor calculated in this way should vanish at the proper temperature equal to Unruh temperature. Since the energy-momentum tensor is normalized with respect to the Minkowski vacuum, such a vanishing is a direct consequence of the Unruh effect—an accelerated medium with Unruh temperature corresponds to the Minkowski vacuum. It is easy to see that energy-momentum tensor (44) satisfies this condition following from the Unruh effect

$$\langle \hat{T}^{\mu\nu} \rangle (T = T_U) = 0 \,. \tag{45}$$

Moreover, as discussed in [26], from the presentation of the result (44) in the form of Sommerfeld integrals, as well as comparison with the field theory in a space with a conical singularity, it follows that the calculated fourth order of perturbation theory is maximal; that is, $\mathcal{O}(a^6) = 0$ at least at $T > T_U$ [26]. Thus, Equation (44) is an exact non-perturbative formula in this region.

We also note that expression (44) can be obtained from the point of view of another approach, where field theory in a space with a conical singularity is considered [25,27]. As discussed in [26], this indicates the duality of the statistical and geometrical approaches to the description of accelerated media.

## 5. Conclusions

The Zubarev density operator provides a powerful fundamental theoretical method for studying quantum-field effects in the accelerated medium. This makes it possible to obtain information about such a medium from the point of view of an inertial observer and there is no need to go to the curvilinear coordinates of the accelerated frame and consider the features of nontrivial space with a boundary. All effects can be calculated in ordinary flat space described by the Minkowski metric using standard Green functions at finite temperature. In this case, the effects of acceleration are calculated in a regular way in the framework of perturbation theory with the boost operator. However, it is possible to obtain exact non-perturbative expressions in the chiral limit, since the first few orders of the perturbation theory are to give a complete perturbative series.

In particular, earlier in [23], the Unruh effect for fermions was demonstrated by calculating fourth-order quantum corrections. In the language of the statistical approach with the Zubarev operator, the Unruh effect should lead to the vanishing of the energy-momentum tensor at the proper temperature equal to the Unruh temperature. Thus, the Zubarev density operator allows one to obtain information about the effects associated with the occurrence of an event horizon in an accelerated system and the radiation associated with it.

In more usual formulation or from the point of view of modern developments in the quantum optics [8–10], the Unruh effect should be manifested in the thermal distribution of photons with Unruh temperature. However, it can be shown that the formula we obtained (44) also contains such a distribution with the Unruh temperature [23].

In this paper, we described the details of the calculations of the coefficients with acceleration in the energy-momentum tensor given in [23], focusing on the calculation of the quantum correction to the energy density. The calculation of this correction consists in finding the mean value of the product of the boost operators and operator of the energy-momentum tensor. Applying Wick theorem, one can transform the average of the product of operators to the product of five thermal propagators. Each of the propagators adds one summation over the Matsubara frequencies and a three-dimensional integral over the momentum, and also each boost operator adds three-dimensional integral over the coordinate and one integral over the imaginary time. The procedure for calculating these sums and integrals is described. In addition, expressions for the coefficients at a finite mass are given.

The effects of acceleration we are discussing are of interest from the experimental point of view, in particular, in heavy-ion collisions, where large acceleration can occur. A systematic study of the effects of acceleration requires calculating the acceleration resulting from particle collisions, similar to calculating the vorticity [34–36]. Since the vorticity turns out to be significant in the collision of particles, acceleration, being another combination of derivatives, is also expected to affect the observables. We predict that the effects of acceleration should be significant at early stages of the collision, when the system is not yet fully thermalized and the terms with acceleration are not suppressed with respect to temperature. In this case, non-equilibrium processes can arise that are associated with instability at the Unruh temperature, which were discussed in [26]. One can also make a prediction that the discussed electron-ion collider (EIC) can become a good laboratory for studying effects of acceleration [37]. An elementary particle like an electron, colliding with an ion, behaves like a wave, which allows us to separate the effects of acceleration from the effects of vorticity.

**Author Contributions:** Conceptualization, G.P., O.T. and V.Z.; investigation, G.P., O.T. and V.Z.; draft preparation, G.P.; supervision, O.T. and V.Z. All authors have read and agreed to the published version of the manuscript.

**Funding:** The work was supported by Russian Science Foundation Grant No 16-12-10059.

**Acknowledgments:** Useful discussions with F. Becattini are gratefully acknowledged.

**Conflicts of Interest:** The authors declare no conflict of interest.

## Appendix A. The Coefficients $a^4$ at Finite Mass

The coefficient $A_2$ at a finite mass is described by the expression

$$
\begin{aligned}
A_2 = & \int_0^\infty d\tilde{p}\,\tilde{p}^2 e^{\frac{9\tilde{E}_p}{2}} \Big( \tilde{E}_p \Big( 9 \left( 51450 - 15619\tilde{m}^2 \right) \tilde{p}^4 + 175 \left( 2450\tilde{m}^2 - 361 \right) \tilde{p}^2 \\
& + 175\tilde{m}^2 \left( 392\tilde{m}^2 - 285 \right) - 140571\tilde{p}^6 \Big) \sinh\left( \frac{\tilde{E}_p}{2} \right) + 27\tilde{E}_p \Big( 27 \left( 53\tilde{m}^2 + 490 \right) \tilde{p}^4 \\
& + 175 \left( 70\tilde{m}^2 - 19 \right) \tilde{p}^2 + 35\tilde{m}^2 \left( 56\tilde{m}^2 - 75 \right) + 1431\tilde{p}^6 \Big) \sinh\left( \frac{3\tilde{E}_p}{2} \right) \\
& - \tilde{E}_p \Big( 9 \left( 247\tilde{m}^2 + 11550 \right) \tilde{p}^4 + 175 \left( 550\tilde{m}^2 + 133 \right) \tilde{p}^2 + 175\tilde{m}^2 \left( 88\tilde{m}^2 + 105 \right) \\
& + 2223\tilde{p}^6 \Big) \sinh\left( \frac{5\tilde{E}_p}{2} \right) + \tilde{E}_p \Big( 9 \left( \tilde{m}^2 + 210 \right) \tilde{p}^4 + 175 \left( 10\tilde{m}^2 + 19 \right) \tilde{p}^2 \\
& + 35\tilde{m}^2 \left( 8\tilde{m}^2 + 75 \right) + 9\tilde{p}^6 \Big) \sinh\left( \frac{7\tilde{E}_p}{2} \right) - 8 \left( \tilde{m}^2 + \tilde{p}^2 \right) \Big( 5\tilde{m}^2 \left( 2\tilde{p}^2 + 63 \right) \\
& + 28\tilde{p}^2 \left( \tilde{p}^2 + 25 \right) \Big) \cosh\left( \frac{7\tilde{E}_p}{2} \right) - 504 \left( \tilde{m}^2 + \tilde{p}^2 \right) \Big( 5\tilde{m}^2 \left( 34\tilde{p}^2 - 9 \right) \\
& + 476\tilde{p}^4 - 100\tilde{p}^2 \Big) \cosh\left( \frac{3\tilde{E}_p}{2} \right) + 56 \left( \tilde{m}^2 + \tilde{p}^2 \right) \Big( 5\tilde{m}^2 \left( 34\tilde{p}^2 + 207 \right) \\
& + 476\tilde{p}^4 + 2300\tilde{p}^2 \Big) \cosh\left( \frac{5\tilde{E}_p}{2} \right) + 1400 \left( \tilde{m}^2 + \tilde{p}^2 \right) \Big( \tilde{m}^2 \left( 70\tilde{p}^2 - 171 \right) \\
& + 4\tilde{p}^2 \left( 49\tilde{p}^2 - 95 \right) \Big) \cosh\left( \frac{\tilde{E}_p}{2} \right) \Big) \big( 50400\pi^2 \left( e^{\tilde{E}_p} + 1 \right)^9 \tilde{E}_p^2 \big)^{-1},
\end{aligned}
\tag{A1}
$$

where the dimensionless quantities $\tilde{m} = m/T$, $\tilde{E}_p = \sqrt{p^2 + m^2}/T$ are introduced. The coefficient $A_4$ at a finite mass has the form

$$
\begin{aligned}
A_4 \;=\; & \int_0^\infty d\tilde{p}\,\Big(\tilde{p}^4 e^{\frac{9\tilde{E}_p}{2}}\Big(1960\sinh\left(\frac{\tilde{E}_p}{2}\right)\cosh^2\left(\frac{\tilde{E}_p}{2}\right)\Big(\big(8\tilde{m}^2+15\big)\cosh\left(2\tilde{E}_p\right) \\
& -8\left(56\tilde{m}^2+15\right)\cosh\left(\tilde{E}_p\right)+984\tilde{m}^2-135\Big)+\tilde{p}^2\left(408170-421713\tilde{p}^2\right)\sinh\left(\frac{\tilde{E}_p}{2}\right) \\
& +27\tilde{p}^2\left(4293\tilde{p}^2+11662\right)\sinh\left(\frac{3\tilde{E}_p}{2}\right)-\tilde{p}^2\left(6669\tilde{p}^2+91630\right)\sinh\left(\frac{5\tilde{E}_p}{2}\right) \\
& +\tilde{p}^2\left(27\tilde{p}^2+1666\right)\sinh\left(\frac{7\tilde{E}_p}{2}\right)+29400\left(14\tilde{p}^2-19\right)\tilde{E}_p\cosh\left(\frac{\tilde{E}_p}{2}\right) \\
& -168\left(2\tilde{p}^2+35\right)\tilde{E}_p\cosh\left(\frac{7\tilde{E}_p}{2}\right)-10584\left(34\tilde{p}^2-5\right)\tilde{E}_p\cosh\left(\frac{3\tilde{E}_p}{2}\right) \\
& +1176\left(34\tilde{p}^2+115\right)\tilde{E}_p\cosh\left(\frac{5\tilde{E}_p}{2}\right)\Big)\Big)\left(1058400\pi^2\left(e^{\tilde{E}_p}+1\right)^9\tilde{E}_p\right)^{-1}.
\end{aligned}
\tag{A2}
$$

The coefficient $A_6$ is zero both for massless and massive Dirac fields

$$
A_6 \;=\; 0\,.
\tag{A3}
$$

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
