# Peer review of "Calculation of Acceleration Effects Using the Zubarev Density Operator"

_2571-712X, doi:10.3390/particles3010001_

Round 1

Reviewer 1 Report

The Unruh effect (like the Hawking effect) is one of the surprising effects to be described by quantum relativistic statistical mechanics. A new and interesting approach has been found by the authors recently and has been published [17, 18] already in Phys. Rev. D. Thus, the general approach is sound and well accepted. The present MS contains previously unpublished calculations, in particular details of the calculations of the coefficients with acceleration in the energy-momentum tensor, new results concerning the higher-order corrections to the energy density, and expressions for particles with finite mass are derived. I estimate this MS as a very interesting application  of the Zubarev method which is well suited to treat this problem. Details of the calculation of quantum correlators are given in the main text and, for finite mass, in the appendix.

Before recommending for publication in MDPI, I have the following suggestions:

Units: The Unruh temperature is \hbar a/(2 \pi c k_B) which may be given at the beginning. At the end of the Introduction the system of units \hbar=c=k_B=1 is defined. p. 1: "This approach is based ..... Zubarev density operator." Here, a reference can be added, e.g. the book of Zubarev (Nonequilibrium Statistical Thermodynamics, 1971) or Zubarev et al. (Statistical Mechanics of Nonequilibrium Processes, 1996) where the Zubarev method of the NSO is worked out. The Conclusions give a summary of the new results obtained in the MS. Is it possible to give a comment whether these results can be seen in experiments, recent or future? The approach is based on a relativistic form (1) of the Zubarev density operator. Is there any relation to other approaches using the relativistic form of the Zubarev density operator such as Theor. Math. Phys. 131, 812 (2002) [arxix:0103021], Theor. Math. Phys. 132, 1026 (2002) [arxix:0106004], Physica A 319, 371 (2003) [arxix:0208083]? The mathematics seems to be sound (I admit that I could not repeat in a short time all calculations). It would be helpful to see which terms and expressions have been already earlier, and which expressions are new, in particular in Sec. 3: Calculations ... and Eq. (43), in particular in comparison with [17, 18].

After considering these comments, I recommending the MS for publication in MDPI.

Author Response

Dear Editor,

We are very grateful to the Referee for the detailed review and interesting suggestions. We took into account all the comments made and integrated the relevant explanations into the paper. Our response and a description of the changes are given below.

Following the proposal of the Referee, we introduce in the Introduction a formula for the Unruh temperature, taking into account fundamental constants.

In the Introduction, we added reference to the book "Nonequilibrium Statistical Thermodynamics, 1971", as a classical monograph, where, in particular, the relativistic density operator, which we use, is introduced.

The effects of acceleration we are discussing are of interest from the experimental point of view, in particular, in heavy-ion collisions, where large acceleration can occur. A systematic study of the effects of acceleration requires calculating the acceleration resulting from particle collisions, similar to calculating the vorticity in [arXiv:1811.00322], [arXiv:1701.00923], [arXiv:1910.01332]. Since the vorticity turns out to be significant in the collision of particles, acceleration, being another combination of derivatives, is also expected to affect the observables. We predict, that the effects of acceleration should be significant at early stages of the collision, when the system is not yet fully thermalized and the terms with acceleration are not suppressed with respect to temperature. In this case, nonequilibrium processes can arise that are associated with instability at the Unruh temperature, which were discussed in [arXiv:1906.03529]. One can also make a prediction that the discussed electron-ion collider (EIC) can become a good laboratory for studying effects of acceleration [PoS DIS 2019, 241 (2019).doi:10.22323/1.352.0241]. An elementary particle like an electron, colliding with an ion, behaves like a wave, which allows us to separate the effects of acceleration from the effects of vorticity. We have added a discussion of the possibilities of experimental verification in the Conclusions.

The method we use is fully consistent with Theor. Math. Phys. 131, 812 (2002) [arxix: 0103021], Theor. Math. Phys. 132, 1026 (2002) [arxix: 0106004], Physica A 319, 371 (2003) [arxix: 0208083], devoted to the peculiarities of consideration of relativistic kinetic theory on three-dimensional hyperplanes. However, in comparison with these works, despite the fact that we are also investigating the relativistic system, we are considering the case of global thermodynamic equilibrium. As was shown in [arXiv:1201.5278], in this case density operator becomes independent on the choice of the three-dimensional hypersurface over which the integration takes place. This allows us to move from the integral over the hypersurface discussed in [arxix: 0103021], [arxix: 0106004], [arxix: 0208083] to the standard hyperplane perpendicular to the time axis, that is, ordinary three-dimensional space-time. Thus, in the case we are considering, we can work without a non-trivial and interesting formalism for constructing a statistical theory on an arbitrary space-like hyperplane. We have introduced the relevant comment in Section 2 after Eq. (3).

Next, we would like to comment on the novelty of the presented results. The method we used to evaluate perturbative in the acceleration terms was developed in a series of works of F. Becattini et al. [arXiv:1505.07760], [arXiv:1704.02808], [arXiv:1807.02071]. We supplemented this technique by a proof of absence of higher order corrections, so that the whole perturbative expansion reduces to a polynomial. In the papers of F. Becattini et al. examples of explicit calculations of corrections of second order were given. As noted in [arXiv:1704.02808], the boost operator does not commute with the Hamiltonian of the system. Because of this, with each subsequent order of perturbation theory, the complexity of calculation of the corresponding quantum correlators increases. The fourth order calculated by us is currently a record one. In this paper, we describe a previously never given scheme for calculating higher orders of the perturbation theory with the boost operator.

Most of the given formulas in Sections 3 and the Appendix, with the exception of the final formulas (42, 43), have not been published anywhere else. In particular, the formulas (22, 23, 28, 29, 31-34, 36-41, A1, A2, A3) are new compared to our previous papers. We have added clarification of the novelty to the Introduction.

We hope that after the changes made, our paper is acceptable for publication in the Particles.

Yours sincerely,
Georgy Prokhorov, Oleg Teryaev and Valentin Zakharov

Reviewer 2 Report

Pls see the report as attached

Author Response

Dear Editor,

We are very grateful to the Referee for the thorough reviewing and valuable comments. We addressed all of them expanding and developing the statements made in our paper.

1. If we compare two approaches: [PNAS 115, 8131 (2018); Phys. Rev. Research 1, 033115 (2019); Phys. Rev. Lett. 91 243004 (2003)] and the relativistic quantum statistical theory, then, in spite of the fact that in both cases the Unruh effect is sufficient, the statement of the problem is different. Approach [Phys. Rev. Research 1, 033115 (2019); Phys. Rev. Lett. 91 243004 (2003)] refers to the field of quantum optics, and the Unruh effect is manifested in the probability of absorption and emission of gamma quanta by atoms, while in the quantum statistical approach, we do not consider explicitly the emission processes, but evaluate mean values of the thermodynamic quantities. In particular, we show that the Minkowski vacuum corresponds to the Unruh temperature, measured by a comoving observer, which is a direct consequence of the Unruh effect from the point of view of quantum statistical mechanics. At the same time, according to [PNAS 115, 8131 (2018)], the Unruh effect manifests itself in the thermal distribution of emitted photons with Unruh temperature.

Despite the difference of the statements of the problem, we can identify a parallel in the two approaches. Indeed, in [Phys. Rev. D 99, 071901(R)] we showed that a term, corresponding to the thermal distribution of gamma quanta at the Unruh temperature, is also presented in the energy density, which we have calculated. It corresponds to the last term in the formula (3.1) in [Phys. Rev. D 99, 071901(R)]). We have added relevant comments on this issue to the Introduction.

2. The perturbation theory considered in the two approaches is different - in [Phys. Rev. Research 1, 033115 (2019); Phys. Rev. Lett. 91 243004 (2003)] the perturbation theory in coupling constant of interaction with an electromagnetic field is considered, while acceleration is taken into account in a nonperturbative way through the Rindler coordinates. And in the case of quantum-field statistical approach we are forced to consider the acceleration effects perturbatively, since they are determined by the boost operator in the density matrix, which does not commute with the Hamiltonian of the system. From this it is clear that the difference in the necessary orders of the perturbation theory in the two approaches is not a contradiction. We have added the relevant comments about the correspondence with [Phys. Rev. Research 1, 033115 (2019); Phys. Rev. Lett. 91 243004 (2003)] to the Introduction and Conclusions.

We hope that after the changes made, our paper is acceptable for publication in the Particles journal.

Yours sincerely,
Georgy Prokhorov, Oleg Teryaev and Valentin Zakharov